# Body Weight Fluctuation as a Risk Factor for Type 2 Diabetes: Results from a Nationwide Cohort Study

**DOI:** 10.3390/jcm8070950

**Published:** 2019-06-30

**Authors:** Kye-Yeung Park, Hwan-Sik Hwang, Kyung-Hwan Cho, Kyungdo Han, Ga Eun Nam, Yang Hyun Kim, Yeongkeun Kwon, Yong-Gyu Park

**Affiliations:** 1Department of Family Medicine, Hanyang University College of Medicine, Seoul 04763, Korea; 2Department of Family Medicine, Korea University College of Medicine, Seoul 02841, Korea; 3Department of Biostatistics, the Catholic University of Korea College of Medicine, Seoul 06591, Korea; 4Department of Medical Statistics, the Catholic University of Korea College of Medicine, Seoul 06591, Korea

**Keywords:** average successive variability, body weight fluctuation, cohort study, weight change, type 2 diabetes

## Abstract

We aimed to investigate how body weight fluctuation affects the risk of developing type 2 diabetes by conducting a nationwide cohort study. A total of 3,855,884 participants from the National Health Insurance System health check-up data from 2012 were included in this study, and follow-up continued until 2016. Body weight was measured at least thrice between 2009 and 2012. Body weight variability (BWV) was estimated using average successive variability (ASV) indices. Cox proportional hazards regression models were used to evaluate the association of BWV with the risk of type 2 diabetes using hazard ratios (HRs) and 95% confidence intervals (CIs). Body weight fluctuation was associated with a higher risk of incident diabetes after adjustment for confounders (HR 1.10, 95% CI 1.07, 1.12 in the highest BWV quartile compared to the lowest). Regardless of the weight change status, the highest ASV quartile of BWV increased the risk for diabetes. Even subjects with a normal glucose tolerance status and those aged under 65 years had a higher risk of diabetes if their body weight highly fluctuated during the follow-up years. Our results suggest that body weight variability is an independent risk factor for diabetes. It is important to pay attention to frequent body weight fluctuations.

## 1. Introduction

Weight fluctuation refers to repeated cycles of weight gain and loss occurring over months or years [1]. It may be due to underlying illnesses or to unsuccessful weight loss maintenance. Studies on repeated weight gain and loss have revealed that such fluctuations cause adverse health outcomes such as stroke, heart disease, bone fractures, and thyroid disease [2,3,4,5,6,7,8,9]. Indeed, highly variable body weights have reportedly been associated with increased cardiometabolic morbidity or mortality [8,9,10,11]. 

There has been a long-standing debate about the association of weight fluctuation with type 2 diabetes [12,13,14,15,16,17,18,19,20,21]. Several early epidemiological studies suggested that weight cycling or weight fluctuation was associated with an increased risk of diabetes [12,13,17,19,21], but the findings from further cohort studies have been inconsistent [14,15,16,18,20]. For example, in a very recent meta-analysis, unstable body weight was significantly associated with the risk of type 2 diabetes [13]. Another recent review by Mackie et al. concluded that weight cycling has no effect on the risk of type 2 diabetes [15]. 

The lack of a standard definition of weight fluctuation, however, may contribute to these inconsistent findings. It is difficult to define weight fluctuation because both the frequency and the amplitude of fluctuation should be taken into consideration [1]. As a result, a wide range of weight fluctuation measures were used in previous studies [12,18,19,20,21]. Only a few studies examined weight variability using average successive variability (ASV) indices in relation to incident diseases [7,12]. Additionally, few studies were based on body weight measurements in a large cohort.

Therefore, the objective of this study was to evaluate body weight fluctuation by measuring average successive variability indices and its association with incident diabetes in a nationwide large-scale population-based cohort derived from the National Health Insurance Service dataset. 

## 2. Materials and Methods

### 2.1. Source of Data and Study Population

The National Health Insurance Service (NHIS) is the universal coverage health insurance service in South Korea. Korea’s insurance system was initiated in 1963 on the basis of the National Medical Insurance Act, which was introduced for companies with over 500 employees in 1977, followed by the achievement of universal coverage in 1989. The NHIS was launched in 2000 as an efficient single-insurer system providing disease prevention, diagnosis, and treatment as well as services related to births, deaths, and health promotion [22]. Accordingly, the NHIS maintains national records for personal information, sociodemographic variables, health care utilization, health screening, and mortality for the population of South Korea. In 2012, the NHIS formed the National Health Information Database (NHID) using information from the existing NHIS database system [23]. The NHID includes data from 2002 to 2016 obtained from a population of over 50 million people who are categorized as insured employees, self-employed insured individuals, or medical aid beneficiaries. The NHID consists of the eligibility database, the national health screening database, the health care utilization database, and the long-term care insurance database. Among these databases, the national health screening database includes information on health behaviors and biomedical variables obtained through the National Health Screening Program (NHSP). The NHSP in Korea has provided free general health check-ups, consisting of various tests and measurements, since 1995. Health screening through the NHSP is available biennially for adults who subscribe to the NHIS [24]. 

We used the national health screening database of adults aged 20 years or older in the whole population of Korea who had been screened in 2012. We recruited 4,365,574 subjects who participated in at least two health check-ups between 2009 and 2011. From this initial population whose body weight was measured at least thrice between 2009 and 2012, we excluded 250,068 subjects identified from the claim database who were prescribed anti-diabetic medication listed under the International Classification of Diseases, 10th Revision (ICD-10) codes E11, E12, E13, or E14 before enrolment. Subjects with diabetes in the 2012 health screening with a fasting plasma glucose (FPG) level of 126 mg/dL or higher (*n* = 116,796) or those with untreated diabetes identified by an FPG level of 126 mg/dL or higher at least once before enrolment (*n* = 142,826) were also excluded. Therefore, 3,855,884 subjects were finally included in the baseline cohort in 2012 and were followed-up until 31 December 2016.

This study was approved by the Institutional Review Board of Hanyang University Hospital (No. 2018-07-044) with permission for the use of data granted from the NHIS (NHIS-2018-1-463). 

### 2.2. Body Weight Variability and Weight Change Status

Systolic and diastolic blood pressure (mmHg), body weight (kg), height (m), and waist circumference (cm) were measured by trained examiners at each health examination between 2009 and 2012. Body mass index (BMI) was calculated as the weight in kilograms divided by the square of height in meters. 

Body weight variability (BWV) was assessed by calculating the ASV indices. The ASV is based on the average absolute difference between successive values. To calculate the ASV in this study, the absolute value of difference between two consecutive measurements of body weight was obtained. Accordingly, BWV was calculated as (D1 + D2 (or + D3))/2 (or 3), where each D is the absolute difference between two consecutive measurements of body weight. The analyses were performed in quartile groups of BWV: <1.00, 1.00–1.66, 1.67–2.65, ≥2.67 kg.

The subjects were categorized into a weight loss group, a weight-stable group, or a weight gain group according to the difference in weight between the two time points: the weight obtained from the initial health check-up in 2009 or 2010 and the weight obtained from the 2012 health check-up. The weight-stable group included the participants whose weight did not change by ±5% between the initial check-up and the 2012 check-up. The weight loss group and the weight gain group consisted of those who lost >5% or gained >5% of their initial weight, respectively. 

### 2.3. Definition of Incident Type 2 Diabetes 

The subjects were followed through 31 December 2016. The diabetes incidence rates were calculated as the number of events per 1000 person-years. The diagnosis of diabetes was indicated by FPG levels of 126 mg/dL or higher from the health check-up database or the presence of claims from the health care utilization database. Using the claim database, individuals with diabetes were categorized as such if they were prescribed anti-diabetic medication listed under the ICD-10 codes E11, E12, E13, or E14 and received a diabetes diagnosis as either principal diagnosis or first to fourth additional diagnosis at least once per year.

### 2.4. Covariate Measurements

Medical histories of cardiovascular disease, stroke, and health-related behaviors such as smoking status, alcohol drinking, and physical activity were obtained through the self-reported questionnaire. History of cardiovascular disease and stroke was assessed using the following question: “Do you have a history of stroke or acute myocardial infarction?” Smoking status was categorized as never, former, or current smoker. Alcohol consumption was categorized based on the frequency of alcohol drinking per week: none, ≤twice/week, or ≥three times/week. Physical activity was classified based on whether the patient participated in strenuous exercise for at least 20 min a week. Household income level was dichotomized at the lower 20%. 

Laboratory measurements, such as serum levels of FPG and cholesterol, were conducted after overnight fasting. The criterion for hypertension was systolic blood pressure ≥140 mmHg or diastolic blood pressure ≥90 mmHg, and serum total cholesterol level ≥ 240 mg/dL indicated dyslipidemia.

### 2.5. Statistical Analysis

The baseline characteristics of the study subjects are presented as numbers and frequencies for categorical variables and as the mean ± standard deviation for continuous variables. For comparisons between groups of ASV quartiles of body weight, the chi-square test was used for categorical variables, and an analysis of variance (ANOVA) was used for continuous variables. 

By dividing the ASV of body weight into quartiles, the association between body weight variability and the risk of new-onset diabetes was estimated with the Cox proportional hazards regression; the results are shown as hazard ratios (HR) and 95% confidence intervals (CI). We ran three models using the hierarchical regression method: (1) a non-adjusted model, (2) a model adjusted for age, sex, smoking status, alcohol consumption, physical activity, income, hypertension, dyslipidemia, and fasting plasma glucose, and (3) a fully adjusted model for the variables in model 2 plus baseline BMI. 

For the subgroup analysis, the ASV indices of body weight were categorized into two classes: the lowest three quartiles for small fluctuation and the highest quartile for large fluctuation. Participants were categorized according to weight change status, FPG level, age, and sex. Participants with large fluctuations of body weight were analyzed for their risk of diabetes compared with those in each group with small fluctuations of body weight.

To examine the association between diabetes risk and body weight fluctuation in the large-fluctuation group according to baseline BMI, the subjects were classified into three groups: a normal group, including subjects with a BMI below 23 kg/m^2^, an overweight group, composed of subjects with a BMI between 23.0 and 24.9 kg/m^2^, and an obese group, with subjects having a BMI of 25 or greater based on the Asia-Pacific guidelines of obesity [25]. The reference group consisted of subjects with normal BMI and small fluctuation of body weight. Cox proportional hazards regression analysis was used to estimate the HR of the highest BWV quartile of each BMI group for new-onset diabetes risk. Statistical analyses were performed using SAS version 9.3 (SAS Institute, Cary, NC, USA).

## 3. Results

### 3.1. Baseline Characteristics

Table 1 presents the baseline characteristics of the participants according to the quartiles of body weight fluctuation. At baseline, the participants who were in the highest quartile were slightly younger. Those in the highest quartile of BWV had a higher body weight and waist circumference at baseline than those in the other three groups. The metabolic parameters and lifestyle characteristics were also significantly different among the quartile groups of body weight fluctuation. Compared to the participants in the three lowest quartiles, a greater proportion of participants in the highest quartile had hypertension, dyslipidemia, and a history of coronary heart disease and stroke.

### 3.2. Risk of New-Onset Diabetes According to Body Weight Fluctuation 

After a median follow-up of 4.4 years, the crude incidences of diabetes were 4.82 per 1000 person-years among the participants with the highest BWV. The highest quartile of BWV was significantly associated with a higher risk of new-onset diabetes after adjusting for baseline BMI and other confounders (HR 1.10, 95% CI 1.07, 1.12) (Table 2). Positive associations were also observed in the analysis of the association between body weight variability and diabetes risk for the variability calculated by standardized deviation, coefficient of variation, and variability independent of mean as well as ASV (Appendix A).

### 3.3. Subgroup Analysis of the Risk of New-Onset Diabetes for Large Body-Weight Fluctuations

Large body-weight fluctuation increased the risk for diabetes in all three groups of weight change status: the weight loss group, the stable-weight group, and the weight gain group. Weight loss of at least 5% of the baseline body weight with large BWV was significantly associated with the risk for diabetes (HR 1.11, 95% CI 1.06, 1.16). Among those whose body weight was stable, participants with the largest BWV had a higher risk of diabetes than those in the lowest three quartiles of BWV (HR 1.07, 95% CI 1.05, 1.10). The risk of diabetes was significantly higher in the large BWV group when body weight increased by more than 5% after adjustment for other variables (HR 1.12, 95% CI 1.08, 1.16). 

In particular, the association between BWV and diabetes risk was significantly stronger in participants whose baseline FPG levels were normal (HR 1.24 95% CI 1.21, 1.27), whereas the association was not significant in those with impaired fasting glucose at baseline (HR 1.00, 95% CI 0.98, 1.02). 

Regardless of age, participants with high BWV had a higher risk of diabetes when divided into a group over 65 years of age and a group under 65 years of age. Finally, both men and women whose weight fluctuated the most had the higher risk of diabetes. Women with large fluctuations of body weight had a higher risk for diabetes than men, with HR of 1.08 for men and 1.14 for women (*p* for interaction, 0.001) (Table 3). 

### 3.4. Risk of New-Onset Diabetes According to Baseline Obesity Status 

Figure 1 shows the incidence rate and HR of new-onset diabetes for the highest quartile of BWV compared with the lowest three quartiles of variability according to baseline BMI divided into three categories. Participants with a BMI of 25 kg/m^2^ or more at baseline with a large fluctuation of body weight in the subsequent years had the highest risk of diabetes compared with those with a BMI less than 23 kg/m^2^ with a small fluctuation of body weight (HR 2.78, 95% CI 2.71, 2.85).

## 4. Discussion 

We observed that body weight fluctuation over 4.4 years was an independent risk factor for new-onset diabetes after adjusting for possible potential confounders, including baseline BMI. Stable weight as well as weight loss and gain of more than 5% were also associated with a higher risk of incident diabetes if the body weight highly fluctuated during this period. Baseline BMI and fluctuation of body weight were associated with an increased risk for diabetes. Moreover, normal glucose tolerance at baseline accompanied by a large fluctuation of body weight during the subsequent years significantly increased the risk of diabetes. Large BWV was associated with a higher diabetes risk in both participants aged under 65 years and those aged 65 years or older. Both men and women had a higher risk of diabetes if their body weight highly fluctuated; the risk was greater for women than for men. 

Our results are in agreement with those of previous studies [5,12,13,19,21,26]. French et al. used root-mean-square error (RMSE), a widely used measurement of body weight variability, to evaluate the relationship between body weight variability and chronic disease incidence in the IOWA Women’s Health Study [21]. A strong association between weight change categories and incident diabetes was found in their study. A study of a middle-aged Finnish population also identified a significant association between weight variability defined as RMSE, measured four times at intervals of one year, and incident diabetes [19]. In a recent report by Rhee et al., body weight variability measured with ASV was found to be associated with an increased risk of diabetes [12]. Another study using ASV as a weight variability measurement demonstrated that the rate of new-onset diabetes increased with each higher quintile of body weight variability in patients with coronary artery disease; in particular, the highest quintile of variability had a 78% higher risk of new-onset diabetes than the lowest quintile [7]. These results are consistent with our findings that measured anthropometric variability by calculating the ASV.

In the present study, if weight change was characterized by large weight fluctuations, weight loss, stable weight, and weight gain were associated with an increased risk of diabetes. In previous studies, weight loss was associated with a decreased risk of diabetes [27,28,29,30], whereas weight gain was associated with an increased risk of diabetes [29,31]. Interestingly, weight loss characterized by large weight fluctuations was revealed to increase the risk of diabetes in our study. Only a few studies have examined weight loss accompanied by large weight fluctuations as one of the risk factors associated with incident diabetes [19,21]. One study demonstrated that the risk of diabetes was twofold higher in participants whose weight fluctuated compared with those with stable weight or moderate fluctuation in a short-term follow-up [21]. Another cross-sectional study examined the association between weight loss and diabetes, demonstrating a relative risk of 1.99 in participants with large body weight fluctuations compared with those who were weight-stable [19]. The results underscore that minimizing weight fluctuation regardless of the weight change status is important for diabetes prevention. However, further studies are needed concerning whether long-term weight loss accompanied by weight fluctuation may lead to the development of diabetes.

Although the mechanism by which body weight fluctuation leads to the development of diabetes is not clear, it seems to be related to hyperinsulinemia and insulin resistance [1]. Weight gain results in the disruption of the metabolic steady state, which indicates a state of hyperglycemia with reactive hyperinsulinemia. If weight loss follows this gain, hyperinsulinemia may be even more pronounced due to the decreased basal metabolic rate [1]. Long-term weight fluctuation interferes with glucose homeostasis, resulting in increased triglyceride and glucose levels. In addition, repeated weight loss and gain may have adverse effects on body fat distribution [26,32,33]. Weight cycling may cause an accumulation of trunk or visceral fat that would explain the increased metabolic risk independent of total adiposity [33]. In one study comparing weight cyclers with non-cyclers, fat cell size and lipoprotein lipase activity were different in the abdomen but not in other parts of the body [32]. A more rapid growth of the adipose tissues and its increased production of leptin, cytokines, and adiponectin could ultimately damage the mechanisms of glucose homeostasis [34].

A comparison of normal glucose tolerance and impaired fasting glucose in evaluating the incident diabetes risk in relation to weight fluctuation was also included in our analysis. In persons with normal glucose tolerance, the risk of diabetes increased even more than in those with impaired fasting glucose, which can be explained by an exaggerated adaptive immune response in the adipose tissue leading to metabolic dysfunction during weight cycling [35]. Studies of the role of hyperinsulinemia have shown that hyperinsulinemia in normoglycemic adults was the strongest predictor of type 2 diabetes [36]. Additionally, this may affect people younger than 65 years of age rather than the elderly in terms of the association between weight fluctuation and diabetes risk. Our results are in line with those of another study that found that weight fluctuation between the ages of 40 and 60 significantly increased diabetes rate (RR = 1.7, *p* < 0.01) [37]. Even non-diabetic middle-aged people should be cautious of the risk of diabetes if they are likely to undergo repetitive weight fluctuation.

However, these results should be interpreted considering several limitations. First, because of the retrospective nature of the study, it is difficult to infer causality. Also, given that the study included at least three observations of body weight measurement throughout four years, different intervals separated the measurements in the study design. Therefore, the results should be carefully interpreted. Second, diabetes incidence could have been underestimated [38]. Information on postprandial glucose and hemoglobin A1c were not available in the NHIS database, and thus, those who might have been diagnosed as having diabetes by high postprandial glucose levels or hemoglobin A1c could be omitted from the data. Third, the study did not examine whether the body weight changes were intentional or unintentional. Unintentional weight change may be attributed to some underlying diseases, but we assumed that body weights of patients with such diseases would not increase again within a relatively short period of time. Fourth, lack of information on dietary habits and educational levels due to lack of data from the NHIS is a further limitation of this study. These factors can have a different effect on the association between weight cycling and diabetes. Fifth, ASV may have been unable to detect body weight variations occurring over a short period of time of less than a year, and the follow-up duration of 4.4 years was relatively short as well. Finally, we could not measure an index of insulin resistance because of the lack of laboratory information, which would have allowed us to better understand the pathophysiologic relationship of anthropometric adiposity variability with the risk of diabetes. However, to the best of our knowledge, this is the first nationwide cohort study exploring the relationship between body weight fluctuation and diabetes incidence. This study also used directly measured anthropometric data rather than self-reported recall data. 

In conclusion, normal and stable body weight maintenance is needed to lower the risk of incident diabetes. This applies to anyone with normal glucose tolerance regardless of sex and age. It is also important to pay attention to weight loss, which can cause weight cycling. Further studies to explore the effect of long-term weight change as well as the variability of obesity parameters on incident diabetes are needed.

## 5. Conclusions

Body weight fluctuations have reportedly been associated with a variety of adverse health outcomes. Our results confirm that large body weight fluctuations are independent risk factors for type 2 diabetes. Large fluctuation of body weight was associated with a higher risk of incident diabetes regardless of weight change status during follow-up periods. Subgroup findings suggested that even subjects with a normal glucose tolerance status and those aged under 65 years at baseline accompanied by large fluctuation of body weight had a higher risk of diabetes. These findings underscore that maintaining a stable body weight is important for diabetes prevention.

## Figures and Tables

**Figure 1 jcm-08-00950-f001:**
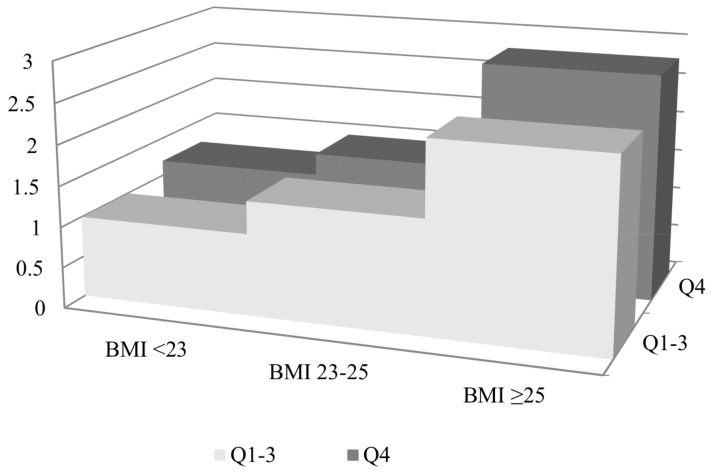
Risk of new-onset diabetes for subjects in the highest quartile of body weight variability compared with those in the lowest three quartiles according to baseline obesity status. Hazard ratio was adjusted for age, sex, smoking status, alcohol consumption, physical activity, income, hypertension, dyslipidemia, and fasting plasma glucose. Black squares: hazard ratio, black bars: incidence rate; Q1–3: the lowest three quartiles of variability; Q4: the highest quartile of variability.

**Table 1 jcm-08-00950-t001:** Baseline characteristics of the study participants according to the quartiles of body weight fluctuation.

	Body Weight Variability (ASV) ^a^				*p* ^b^
	Q1	Q2	Q3	Q4	
*n*	1,125,230	909,697	838,844	982,113	
Age (years)	45.62 ± 11.25	44.69 ± 11.02	43.99 ± 11.62	41.32 ± 11.51	<0.001
Sex (male, %)	686,313 (60.99)	617,197 (67.85)	577,603 (68.86)	704,187 (71.7)	<0.001
Height (cm) ^c^	165.41 ± 8.6	166.73 ± 8.54	167.3 ± 8.64	168.79 ± 8.57	<0.001
Body weight (kg) ^c^	63.15 ± 10.99	65.36 ± 11.23	66.79 ± 11.57	70.46 ± 12.71	<0.001
Waist circumference (cm) ^c^	78.21 ± 8.66	79.52 ± 8.57	80.43 ± 8.65	82.44 ± 9.1	<0.001
BMI (kg/m^2^) ^c^	23.02 ± 2.93	23.45 ± 2.97	23.81 ± 3.06	24.67 ± 3.4	<0.001
Smoking status					<0.001
Non-smoker	494,831 (44.05)	366,189 (40.32)	335,965 (40.11)	378,029 (38.55)	
Ex-smoker	549,898 (48.95)	470,391 (51.8)	432,733 (51.67)	514,782 (52.49)	
Current smoker	78,579 (7)	71,596 (7.88)	68,870 (8.22)	87,830 (8.96)	
Alcohol consumption					<0.001
None	630,177 (56.05)	459,670 (50.56)	413,663 (49.35)	453,304 (46.18)	
≤Twice/week	198,285 (17.64)	177,984 (19.58)	162,727 (19.41)	192,835 (19.65)	
≥Three times/week	295,842 (26.31)	271,424 (29.86)	261,909 (31.24)	335,421 (34.17)	
Physical activity (regular)	234,258 (20.83)	193,048 (21.23)	176,553 (21.06)	212,629 (21.66)	<0.001
Household income (lower 20%)	191,579 (17.03)	144,372 (15.87)	138,953 (16.56)	153,461 (15.63)	<0.001
Hypertension ^d^	206,915 (18.39)	168,887 (18.57)	159,818 (19.05)	180,666 (18.4)	<0.001
Dyslipidemia ^d^	175,343 (15.58)	141,105 (15.51)	133,160 (15.87)	152,793 (15.56)	<0.001
History of heart disease	9807 (1.16)	7550 (1.08)	7750 (1.22)	8176 (1.09)	<0.001
History of stroke	3007 (0.36)	2344 (0.34)	2688 (0.42)	3079 (0.41)	<0.001
Systolic blood pressure (mmHg)	120.25 ± 13.69	120.91 ± 13.55	121.34 ± 13.58	121.9 ± 13.48	<0.001
Diastolic blood pressure (mmHg)	75.62 ± 9.54	76.13 ± 9.48	76.31 ± 9.5	76.7 ± 9.49	<0.001
Fasting plasma glucose (mg/dL)	92.42 ± 10.73	92.61 ± 10.73	92.64 ± 10.75	92.48 ± 10.84	<0.001
Total cholesterol (mg/dL)	194.26 ± 34.61	194.87 ± 34.57	195.03 ± 34.91	195.28 ± 35.56	<0.001

Data are expressed as the mean ± standard deviation for continuous variables and as number (%) for categorical variables. ^a^ Q1, <1.00 kg; Q2, 1.00–1.66 kg; Q3, 1.67–2.65 kg; Q4, ≥2.67 kg; ^b^
*p* values were obtained by using the analysis of variance for continuous variables and the chi-square test for categorical variables; ^c^ measured in 2012 health check-ups as the point at which follow-up started; ^d^ The criterion for hypertension was systolic blood pressure ≥140 mmHg or diastolic blood pressure ≥90 mmHg, and serum total cholesterol level ≥240 mg/dL indicated dyslipidemia. ASV, average successive variability; Q, quartiles; BMI, body mass index.

**Table 2 jcm-08-00950-t002:** Risk of new-onset diabetes according to body weight fluctuations.

	Group	*n*	Incident Diabetes	Duration	Incidence ^a^	Model 1 ^b^	Model 2 ^c^	Model 3 ^d^
Body weight variability (ASV) ^e^	Q1	1,125,230	20,793	4,906,440.78	4.24	1.00	1.00	1.00
	Q2	909,697	16,652	3,966,357.77	4.20	1.04 (1.02, 1.06)	1.02 (1.001, 1.04)	0.99 (0.97, 1.01)
	Q3	838,844	16,384	3,647,718.23	4.49	1.13 (1.10, 1.15)	1.09 (1.06, 1.11)	1.01 (0.99, 1.03)
	Q4	982,113	20,546	4,262,509.48	4.82	1.39 (1.36, 1.41)	1.29 (1.27, 1.32)	1.10 (1.07, 1.12)

^a^ Incidence rate per 1000 person-years; ^b^ Model 1 was non-adjusted; ^c^ Model 2 was adjusted for age, sex, smoking status, alcohol consumption, physical activity, income, hypertension, dyslipidemia, and fasting plasma glucose; ^d^ Model 3 was adjusted for the variables in model 2 plus baseline BMI ^e^ Q1, <1.00 kg; Q2, 1.00–1.66 kg; Q3, 1.67–2.65 kg; Q4, ≥2.67 kg.

**Table 3 jcm-08-00950-t003:** Subgroup analysis of the risk of new-onset diabetes for subjects in the highest quartile of body weight variability compared with those in the lowest three quartiles of variability.

	Highest Quartile of BWV	
	HR (95% CI) ^a^	*p* for Interaction
Weight change status (%)		0.43
Loss (≥5)	1.11 (1.06, 1.16)	
Stable (±<5)	1.07 (1.05, 1.10)	
Gain (≥5)	1.12 (1.08, 1.16)	
Fasting plasma glucose (mg/dL)		<0.001
<100	1.24 (1.21, 1.27)	
≥100, <126	1.00 (0.98, 1.02)	
Age (years)		0.19
<65	1.10 (1.08, 1.12)	
≥65	1.09 (1.04, 1.14)	
Sex		0.001
Male	1.08 (1.06,1.10)	
Female	1.14 (1.10,1.17)	

^a^ Adjusted for age, sex, smoking status, alcohol consumption, physical activity, income, hypertension, dyslipidemia, fasting plasma glucose, and BMI at baseline; BWV, body weight variability; HR, hazard ratio; CI, confidence interval.

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
