# Peer review of "Body Weight Fluctuation as a Risk Factor for Type 2 Diabetes: Results from a Nationwide Cohort Study"

_jcm, 2019, doi:10.3390/jcm8070950_

Reviewer 1 Report

The manuscript describes the rationale, methods and results of the effect of weight variability in people with type 2 diabetes.

The background is sufficient, methods are adequately described, and conclusions are supported by the results. Importantly authors have descibed the limitations of this work, which contributes to the knowledge of the specific domain.

I have only one further comment and it is with respect to table 1. I cannot understand how some of the vairables on each quartile yield a significant difference (<0.001) when it is obvious that they are overlaped (Age, Height, Waist, Fasting plasma glucose, systolic and diastolic BP....)

Author Response

     We are grateful of your thoughtful review and kind words for our paper. We understand your perspective that some values of each quartile appear to be overlapped. As we checked again, this analysis was carried out correctly. The results of significant difference of these variables are the results of the analysis of variance (ANOVA) of the mean values of the four quartile groups. Therefore, when one of the four values shows a difference compared to the other three, a p-value result of significant difference can be obtained. Also, the large number of N results in a p-value of<0.001 being calculated. Given that Table 1 was intended to give a brief description of the baseline characteristics of the participants who were in the highest quartile, we would appreciate your understanding that we did not do post-hoc analysis.
    We found the wrong expression in the Table1 annotation, by the way, and we fixed it. The expression 'independent t-test for continuous variables' is incorrect and should be changed to "analysis of variance for continuous variables". This expression is correctly described in the main text, which is line 130, page 3 of the Statistical analysis section, though. Therefore, we modified the annotation as follows:

Line 162, page 4, Result section)

        P values were obtained by using the independent t-test analysis of variance for continuous         variables and the chi-square test for categorical variables
We again appreciate the kindness of the Editor and Reviewers in helping improve the manuscript.

Yours Sincerely,
Hwan-Sik Hwang

Reviewer 2 Report

These are my general/specific comments.

In their paper entitled “Body weight fluctuations as risk factors for type 2 diabetes: Results from a nationwide cohort study”, Park et al. evaluated body weight fluctuation by measuring average successive variability indices and its association with incident diabetes via a nationwide large-scale population-based cohort derived from the National Health Insurance Service dataset. 

Comments: 
After revision, this paper, mainly the discussion, was improved. 

Author Response

 We thank the reviewer for thoughtful review of our work and kind words. Thank you very much.